# Neurodevelopmental Disorders and Suicide: A Narrative Review

**DOI:** 10.3390/jcm13061627

**Published:** 2024-03-12

**Authors:** Antonella Gagliano, Carola Costanza, Irene Di Modica, Sara Carucci, Federica Donno, Eva Germanò, Costanza Scaffidi Abbate, Michele Roccella, Luigi Vetri

**Affiliations:** 1Department of Human and Pediatric Pathology “Gaetano Barresi”, University of Messina, 98125 Messina, Italy; antonellagagliano.npi@gmail.com (A.G.); dmdrni95s67l750k@studenti.unime.it (I.D.M.); eva.germano@unime.it (E.G.); 2Department of Sciences for Health Promotion and Mother and Child Care “G. D’Alessandro”, University of Palermo, 90128 Palermo, Italy; 3Department of Biomedical Sciences, Section of Neuroscience & Clinical Pharmacology, University of Cagliari, 09124 Cagliari, Italy; sara.carucci@gmail.com (S.C.); federica.donno87@gmail.com (F.D.); 4Child & Adolescent Neuropsychiatry Unit, “A. Cao” Paediatric Hospital, 09121 Cagliari, Italy; 5Department of Psychology, Educational Science and Human Movement, University of Palermo, 90128 Palermo, Italy; costanza.scaffidi@unipa.it (C.S.A.); michele.roccella@unipa.it (M.R.); 6Oasi Research Institute-IRCCS, Via Conte Ruggero 73, 94018 Troina, Italy; lvetri@oasi.en.it

**Keywords:** suicide, neurodevelopmental disorders, pediatric population

## Abstract

Specific risk factors for self-harm and suicide in children and adolescents with neurodevelopmental disorders (NDD) may differ from those in the general population within this age range. In the present review paper, we conducted a narrative analysis of the literature, aiming to establish a connection between suicide and affective disorders in children and adolescents with NDD. Emotion dysregulation (ED) as an individual factor and adverse childhood experiences (ACE) as environmental factors are discussed as risk factors for suicidality in all individuals with NDD. We propose a theoretical model in which ED and ACE can directly lead to self-harm or suicide, directly or indirectly by interacting with depressive spectrum disorders. Additionally, we suggest that specific risk factors are more frequently associated with each of the neurodevelopmental disorders listed in the DSM-V. This review underlines the key points useful to improve the knowledge of the trajectory leading to suicide risk in NDDs with the purpose to facilitate the early identification of the suicide risk.

## 1. Introduction

Every year, all around the world, more than 700,000 people take their own lives. Each of these deaths is a tragedy, as they could be avoided with the appropriate prevention strategies.

Suicide rates vary widely between countries. The global age-standardized suicide rate for 2016 was 10.5 per 100,000. However, rates varied widely between countries, from 5 suicide deaths per 100,000 to over 30 per 100,000 [1]. It is essential to acknowledge that data on the number of suicides are often significantly underestimated due to social and cultural stigma and its legal consequences. Moreover, suicide was the second highest cause of death among young people aged between 15 and 29, surpassed only by road accidents [2]. Suicidal spectrum behaviors (SSBs) include a continuum that encompasses suicidal ideations, suicidal attempts, suicidal plans, and ultimately completed suicides.

Scientific research aimed at understanding the causative factors determining SSBs represents the fundamental premise for implementing appropriate and more efficient strategies for preventing this phenomenon. There are many risk factors that increase suicide risk, including bullying, financial distress, and trauma, but mental illnesses represent the major risk factor [3].

Children and adolescents with neurodevelopmental disorders (NDDs) show a wide range of self-destructive behaviors ranging from non-suicidal self-injury (NSSI, defined as the intentional damage of the body tissue without clear suicidal intent), to suicidal attempt (SA, defined as non-fatal self-injurious behavior with the intent to die), until suicide death [4].

However, suicidal ideation and gestures have been, in past decades, greatly under-reported in the pediatric and adolescent population and definitely poorly studied. The Center for Disease Control and Prevention survey data reported that 14% of high school students seriously considered killing themselves, 6% attempted suicide in the previous year, and 2% required medical attention [5,6]. However, suicidality within the pediatric population has been more often reported in adolescents diagnosed with psychotic disorders, mostly in the presence of positive symptoms, depression, and anxiety disorders [7]. Very high risk was also found in young patients with bipolar disorder, specifically in bipolar disorder with psychotic symptoms [8], panic, and conduct and oppositional defiant disorder [9].

Suicidal spectrum behaviors have been less often described in the pediatric population with NDDs, such as autism or intellectual disability, especially in severely/profoundly mentally retarded patients [10]. More recently, NDDs have been seriously taken into consideration as risk factors for suicidality consistent with the psychopathology gradually emerging across development and progressively increasing the risk of SSB. 

The effect of NDDs on suicidal risk seems to be directly due to the neuropsychological and functional impairment. Additionally, there is an indirect link to adverse childhood experiences (ACE) presenting with a greater frequency in individuals with NDDs than in the neurotypical development population. The current state of knowledge reports that emotion dysregulation plays a core function in the development of ideas and behaviors of self-inflicted violence, such as NSSI and suicidal ideation. In particular, the negative valence system (NVS) has been primarily studied as responsible for responses to aversive situations or contexts, such as fear, anxiety, and loss. When individuals are easily overwhelmed by negative emotions, they may become more at risk for suicidal desire and the acquisition of capability for suicide (ACS) [11,12]. Emotional dysregulation is a key variable to be considered within the framework of the interpersonal theory of suicide that is a comprehensive theory about the proximal, jointly necessary, and sufficient causes for suicidal behavior identifying the main causal factors for suicide, such as perceived burdensomeness, failed belongingness, and acquired capability for suicide [13,14,15]. 

According to this paradigm, there is evidence that universal suicide risk screening implementation in pediatric NDDs in order to identify increased suicide risk and referral to mental health care is feasible and potentially extremely useful, although the high rates of declined participation that previous experiences have highlighted further underline how great the social stigma is related to the suicidal phenomenon [16].

Starting from this conceptual frame, we revised the literature contributions on suicide and suicide behaviors in children and adolescents with NDDs, describing the available data for each of the six disorders included in the DSM-5 chapter on NDDs. The review intends to summarize the key points useful to improve the knowledge of the trajectory leading to suicide risk in NDDs. The purpose is to increase the attention on this issue in clinical practice in order to facilitate the early identification of the suicide risk and, hopefully, the prevention of this tragic outcome.

## 2. Materials and Methods

### 2.1. Search Strategy

The relevant literature was obtained from the biomedical database, without limiting the search period to capture a broad range of potential studies. The search was carried out until 30 September 2023. This review intends to present the available research about suicide in individuals with NDDs. The databases PubMed and Google Scholar were used to search the following terms “suicide” or “non-suicidal self-injury”, or “NSSI”, combined with NDDs (DSM-5): “Intellectual Disability”, “Communication Disorders”, “ADHD”, “Autism Spectrum Disorder”, “Specific Learning Disorder”, “Tic” and “Tourette disorder”. Various forms of suicidal outcomes were considered, including suicidal ideation (SI), suicidal planning (SP), non-suicidal self-injury (NSSI), deliberate self-harm (DSH), suicide attempt (SA), and suicide or suicidality. The search was carried out by applying the following syntax: (suicide OR non-suicidal self-injury) and (neurodevelopmental disorders OR intellectual disability OR ADHD OR autism OR specific learning disorder OR tic OR Tourette disorder). English-written full-text clinical studies, case reports, and reviews were included. Exclusion criteria included textbooks, editorials, letters to the editor, and contents not connected to the topic of our review. Additional literature was obtained by searching the manuscripts’ references (snowball method). A framework with six categories, one per NDD, was predefined. 

### 2.2. Data Extraction and Synthesis

The results of the search process are summarized in Figure 1. Out of a total of 444 papers selected, 124 duplicates were removed. A total of 320 records were screened, and 132 were considered relevant to be included in this review and remained for full-text screening. Finally, 127 articles were included in this review. 

## 3. Suicide in the NDD 

### 3.1. Intellectual Disability and Suicide 

Diagnostic and Statistical Manual of Mental Disorders 5th Edition (DSM-5) defines intellectual disability (ID) as a neurodevelopmental disorder that begins in childhood and is characterized by intellectual difficulties as well as difficulties in conceptual, social, and practical areas of living [17]. The DSM-5 emphasizes the need to use both clinical assessment and standardized testing of intelligence when diagnosing intellectual disability, with the severity of impairment based on adaptive functioning rather than IQ test scores alone [18].

Studies have shown that 30–64% of children and adolescents with ID develop comorbid mental health disorders, such as depression, anxiety, and psychosis. This is a rate 2.8–4.5 times higher than their peers [19]. The presence of psychiatric disorders and behavioral problems in individuals with intellectual disabilities is also higher than in the general population (approximately four times higher) [20,21]. However, suicidal behavior in people with intellectual disabilities (IDs) has received little attention in the scientific literature. Much of this limited research has been based on retrospective case study approaches or empirical data analysis, with minimal direct involvement from either individuals with IDs or disability support workers [22].

People with intellectual disabilities (IDs) are more likely to be exposed to risk factors for suicide, such as mental health issues, higher unemployment, lower education, and previous self-harm or suicide attempts [23]. Compared to people without disabilities, youths with IDs have been reported to experience more abuse, neglect, social disadvantage, challenging family circumstances, stigma, and peer exclusion [24]. They also experience depression at a higher rate [25]. Psychopathological symptoms in individuals with intellectual disabilities are exacerbated by a condition of vulnerability and inadequate cognitive and emotional tools that impair environmental adaptation, though in different ways depending on the severity of intellectual disability and individual functioning [25]. 

For a long time, in some areas of healthcare, it was believed that the presence of IDs could act as a protective factor against suicide, maybe due to the lack of cognitive sophistication to conceptualize, plan, or carry out suicide [26,27,28]. Thus, suicide attempts in ID subjects were considered comparable to those of the general population (Hurley, 2002), as well as the techniques chosen to commit suicide, including choking on objects, ingesting poisonous liquids [29], jumping out of windows or under cars, overdosing on drugs, shooting, stabbing, and slitting one’s wrists [30]. However, in the recent literature, individuals with IDs have been recognized as being both capable of forming intent for suicide and acting on this intent [31]. In particular, suicidal thoughts and attempts seem to be more common in people with mild intellectual disabilities and less frequent in people with severe retardation and profound mental disability [25]. 

There are no suicide screening tools designed especially for young people with intellectual disabilities [32]. A suicide risk screening instrument specifically designed to evaluate children and adolescents with IDs would greatly aid clinicians in a variety of settings. While such a measure is perhaps best utilized in a group with an IQ in the mild to moderate range (IQ 75–45), clearly more advanced expressive/verbal abilities make it easier to assess the child [32].

Priority actions not only involve the accurate diagnosis and treatment of mental health conditions associated with IDs (particularly depression and symptoms more closely associated with suicide risk), but they also severely limit the access and availability of extremely lethal suicide means (such as ropes, guns, gases, toxic substances, alcohol, and drugs), which should not be present in the lives of individuals with intellectual disabilities [25]. Moreover, the screening process should include informant components from teachers and caregivers to collectively assess current and past suicidal ideation and behavior, including any past attempts. Parent observation of troubling behavior changes or regression of functional skill level (over the past 2 to 3 months) would also be an important component [33].

### 3.2. Communication Disorders and Suicide 

Communication disorders (CD) involve difficulties in the development of language, speech, and communication skills. Language disorders involve difficulties in the acquisition and use of language across three functional levels (speech, language, and social or pragmatic communication), with onset in the early developmental period [18]. Compromises can have varying levels of severity and can coexist with each other [34]. CD is one of the most common neurodevelopmental disorders currently diagnosed, with an estimated prevalence of 7.58% [35]. Early diagnosis and recognition in childhood can lead to improved outcomes with treatment [36]. Without timely intervention, CD may interfere with relational aspects, communication, and norms of social behavior [37].

We found no studies that specifically investigated the possible correlation between language developmental disorders and suicidality. In a study of 233 patients, which diagnosed several cases of developmental speech and language disorders using DSM-III-R criteria, no statistically significant correlation with suicidal ideation/attempts was demonstrated [10]. Despite the lack of studies on suicidality, emotional and behavioral problems in children diagnosed with a CD have been widely described in the past, also highlighting the high risk of progression towards psychiatric disorders. Much research has shown that adolescents with language development disorder (DLD) may have several problems with peers [36], such as an increased sense of loneliness and a tendency towards isolation. At 9 years old, children with CDs have a lower perception of their quality of life and social-emotional problems [37]. Childhood language disorder is also a specific risk factor for social anxiety because of the typical characteristic of challenging communication with others [38]. Developmental language disorders were also associated with higher depressive symptoms compared to the general pediatric population [39].

In light of these aspects, several correlations with the main risk factors for suicidal behavior and suicide attempts should be taken into consideration as potential conditions leading to suicide also in individuals with CD. With this purpose, it has been suggested to highlight the emotional and behavioral symptoms on which a network of experts and language therapists can work simultaneously [40]. Some authors underline the difficulties of preventing negative emotional outcomes in individuals with CDs due to a paucity of research conducted on this disorder relative to other neurodevelopmental disorders [41].

An interesting conceptual frame is that CDs can also affect the capability to make conversations about emotions and, with this skill, improve the emotion regulation capability. The causal role of language in the development of emotion regulation suggests that targeting language as a malleable target for intervention could improve emotion regulation skills in children with neurological disorders (ND) [42].

### 3.3. ADHD and Suicide 

Individuals with attention-deficit/hyperactivity disorder (ADHD) present an increased risk of facing mental health challenges, social difficulties, and premature mortality during the transition into adulthood [43] with more frequent visits to the emergency rooms (odds ratio 1.93, 95%CI: 1.35, 2.74; *p* < 0.001; [44]) and a more than twofold higher likelihood of mortality compared to those without ADHD [45].

Evans, Hale et al. [46], in their retrospective study including the records of patients presenting at the emergency departments (ED), showed that individuals without ADHD and with different ADHD subtypes (overall ADHD, inattentive, hyperactive, and combined) differ in the frequency of all types of injuries (fractures, accidental overdose, burn injury, drowning incident, gunshot wounds, suffocation) as well as suicide attempts. Subjects without ADHD were less than 1/6th as likely to attempt suicide compared to the overall ADHD cohort and all ADHD subtypes were slightly less likely to attempt suicide than the non-ADHD cohort (hyperactive: OR = 5.84, *p* < 0.001; inattentive: OR = 6.27, *p* < 0.001; combined: OR = 5.84, *p* = 0.031).

The association between ADHD and SSB remains uncertain, with contrasting findings in the literature. While some studies have reported a significant correlation between ADHD and suicidal ideations, attempts, or completed suicides [47,48,49], some others did not confirm the same results [50,51]. A meta-analysis including 57 studies with a total of 90,805 participants diagnosed with ADHD and 239,778 without ADHD revealed a significant association between ADHD and suicidal attempts (OR 2.37, 95% CI = 1.64–3.43; I^2^ = 98.21), suicidal ideations (OR 3.53, 95% CI 2.94–4.25; I^2^ = 73.73), suicidal plans (OR 4.54, 95% CI 2.46–8.37; I^2^ = 0), and completed suicide (OR 6.69, 95% CI 3.24–17.39; I^2^ = 87.53), with a robust association when adjusting for possible confounders and moderators [52].

A recent study by Trivedi et al. [53] based on the National Inpatient Hospitalization data indicated that the prevalence of suicidal ideation was 25.1% in adolescents with ADHD compared to the 10.3% of a control non-ADHD group. Similarly, the incidence of suicidal attempts was higher in the ADHD group (8%), as opposed to 3.9% in the control group. After controlling for covariates, ADHD emerged as a robust predictor of suicidal ideation/attempt, with an odds ratio of 2.18, supporting the high association between ADHD and the elevated risk of suicidality in adolescents. While several findings suggest that ADHD may be a risk factor for SSBs, further research is warranted to fully explain the extent to which ADHD independently contributes to an increased risk of suicidal spectrum behaviors.

Taking into account that ADHD subtypes can change over time, the research focused on examining the relationship between ADHD symptom severity and SSBs and revealed the combined subtype as the higher risk factor for self-harm behaviors [54], suicidal ideation [54,55], and suicide attempt [48]. In terms of symptom severity, both inattention and hyperactivity/impulsivity symptom severity scores represent childhood predictors of NSSI and SA [54], supporting the hypothesis that overall ADHD symptom severity is an important factor in relation to SSBs. Additionally, persistency of the symptoms from childhood into adolescence seems to be associated with more frequent SSBs [56]; females with diagnostic persistence from childhood to adulthood showed significantly higher risks of both NSSI and SA when compared to non-ADHD (OR 6.0 and 5.8) or ADHD subjects with an age-related decline in ADHD symptoms (OR 6.1 and OR 10.6).

The association between ADHD symptoms and SSBs could reflect the critical role played by challenges associated with hyperactivity, inattention, and impulsivity. For example, the presence of both inattentive and hyperactive/impulsive symptoms may increase the risk factors for suicide in youths, such as social and familial conflicts [57]. Youths with ADHD are known to experience social difficulties with peers and have a heightened risk of mental health struggles, such as depression [58], which may contribute to the observed differences in the likelihood of suicide attempts between patients with and without ADHD.

The recent systematic review by Austgulen et al. [59] showed that several demographic and clinical features are associated with an increased risk of SSBs in adolescents and adults with ADHD. The rate of suicide attempts seems to be higher in females (ratio: 1:4) than in males (ratio 1:7), with females presenting a higher prevalence of associated mood disorders [60].

Executive functioning and impulsivity are considered significant contributors to the risk of self-harm and suicidal behaviors acting both as mediating and predictive factors [61,62]. The meta-analysis conducted by Liu et al. in 2017 indicated distinctive associations between behavioral impulsivity and cognitive impulsivity [63,64,65] within a spectrum of suicidal behaviors. Specifically, deficits in behavioral impulsivity (i.e., inhibiting the initiation of a behavior or stopping a behavior) were linked to NSSI, while both behavioral and cognitive impulsivity (impulsive choice, as the tendency to prefer small immediate rewards over larger delayed ones) were associated with suicide attempts. Notably, a more robust relationship was observed between behavioral impulsivity and suicide attempts within the past month compared to lifetime attempts [66].

Executive functions and impulsivity are also related to self-regulatory processes and emotional dysregulation [67], two core components of ADHD. It is likely that incorporating measures of these functions would help in estimating the SSB risk. Dimensions of emotional dysregulation represent a significant risk factor for SSBs [59], increasing symptom severity in ADHD and contributing to a higher prevalence of comorbid disorders and challenges in daily life activities. The different symptom presentation and the association with comorbid disorders, therefore, reinforce the hypothesis that emotional dysregulation serves as a significant potential mediating factor, contributing to an elevated risk of SSBs in adolescents with ADHD.

While individuals with ADHD exhibit a decreased life expectancy and more than double the risk of death compared to individuals without ADHD [44], the research on the impact of psychiatric comorbidities on determining a higher or lower risk of SSBs yielded conflicting data [68,69], failing to quantitatively assess the impact of their possible moderating role. Some researchers concluded that the relationship between ADHD and SSBs [70] is influenced by the cumulative effects of externalizing substance use disorders, affective disorders (emotional problems in males and depression in females), or somatic conditions [71,72,73,74,75,76], rather than being attributed to ADHD only.

Wiener et al. (2019) indicated that the presence of comorbid oppositional defiant disorder, conduct disorder, and substance use disorder heightened the risk of death in individuals with ADHD [77]. However, excluding individuals with these comorbid conditions, the mortality rate still increased by 50% in those diagnosed with ADHD, compared to those without ADHD (MRR 1·50, 1·11–1·98), supporting that the higher mortality linked to ADHD was only partially attributable to these comorbid conditions. Furthermore, women with ADHD without oppositional defiant disorder, conduct disorder, or substance use disorder had a 2.85 higher risk of death compared to their counterparts without ADHD, whereas in boys and men, the risk was 1.27 (0.89–1.76) [45,78].

Other findings support the hypothesis that psychiatric comorbidities are “confounding but not responsible” for the ADHD–SSB relationship [52,59]. However, some results suggest that the presence of comorbid externalizing disorders, such as substance use disorders (SUDs), and internalizing disorders, such as depression and anxiety, heightens the risk of SSBs in individuals with ADHD [59].

Stimulant and non-stimulant medications constitute the primary therapeutic approach and have demonstrated efficacy in alleviating symptoms of ADHD [79]. The relationship between ADHD medication and suicide attempts has been the subject of investigation in numerous epidemiological studies [78,80,81,82,83], with a lack of consistent results. Some evidence supports a protective effect of stimulant medications, indicating a lower risk of suicide attempts in youths with ADHD who received methylphenidate (MPH) [84], particularly after 90 days from the initial intake, and a more pronounced reduction after more than 180 days of use [85].

A population-based electronic medical record study [86] confirmed the protective role of stimulants. Among users of non-stimulant/mixed medications, no significant increase in the rate of suicide-related events was observed. The incidence of suicide attempts was, in fact, higher in the period immediately before the initiation of MPH treatment, and although the risk remained elevated right after starting MPH treatment, it returned to baseline levels during the continuation of the treatment. Non-stimulant treatment did not demonstrate an increased risk of suicide attempts [86,87]. Nevertheless, patients undergoing ADHD medication should be closely monitored for the emergence of psychotic symptoms, depression, irritability, and suicidal ideation as part of routine care.

### 3.4. Autism Spectrum Disorder and Suicide 

Among neurodevelopmental disorders, autism spectrum disorder (ASD) has a net prevalence worldwide of about 1/100 [18], with an increasing trend [88]. It is characterized by persistent communication and social interaction difficulties and restricted and repetitive behaviors, which can vary in severity. 

Currently, few population studies have determined the incidence [89] of suicide deaths in ASD patients [90]. One of the first retrospective studies on this issue reported that young patients with ASD were less likely to engage in SSBs than young people with other psychiatric diagnoses [10]. Recently, however, several studies attest to a higher overall mortality rate of ASD patients compared to the general population, some even double, describing suicide as one of the potential causes of death [91]. Self-injurious behavior, suicidal thoughts, and suicide attempts are also described as more common in ASD patients than in the general population [92,93]. A study conducted in Utah over 20 years affirmed that young people with ASD were at over twice the risk of suicide than young people without ASD (0.11%; *p* < 0.05) [94]. Interestingly, the higher prevalence of suicide in the ASD population described by Kirby and colleagues was driven by the female risk rate that was over three times higher than in the female general population (relative risk (RR):3.42; *p* < 0.01).

Several factors have been described to be related to an increased risk of suicidal behavior in ASD people. A Swedish population-based register study analyzed the risk of SSB in patients and families with ASD [95]. A sample of 54,168 individuals was recorded from 1987 to 2013. Among this population, the risk of SSBs was highest in ASD females without intellectual disability and with ADHD, compared to general population controls. Approximately 8–12.5% of individuals in the ASD-without-ID group had attempted suicide at some point during life [95]. Furthermore, higher levels of autistic traits may frequently be detected in adults who have attempted suicide, even though no particular temperamental or character traits related to suicidal ideation or attempts have been seen in adults with ASD [96].

An increased risk of self-injurious behavior has been described in younger ASD patients with intellectual impairment, while there is a higher risk of suicide in those with a higher intelligence quotient (IQ) [97]. Those who were found to have more significant deficits in social communication also had a higher risk of suicide attempts, suicidal planning, and ideation [98] but not self-harm without suicidal intent. Several findings suggest that social impairments in establishing interpersonal relationships are triggers for SSBs [92,99] along with the deficit in communicating and understanding feelings and thoughts [89]. Both these conditions interfere with the opportunity to share their negative emotions and to elaborate coping strategies. In typically developed individuals, solid social relationships count as a protective factor in the context of depressive symptoms, but social communication difficulties due to autistic traits represent an inherent risk for suicidality [97].

Adolescents with ASD without intellectual disabilities are at risk for suicidal behavior due to increased awareness of their communication problems and secondary depression associated with social isolation and exclusion [99]. The risk of contemplating suicide [100] ideas and dying by suicide [95] is the highest for autistic adults without intellectual disabilities (IDs), especially in those with a significantly higher level of autistic traits [100].

People with autism are also four times more likely to develop depression than the general population [101]. However, it is difficult to recognize symptoms in people with ASD, in whom low mood is generally expressed through restlessness and insomnia rather than the expression of feelings of sadness. Adults with ASD have instead reported increased levels of sensory discomfort or changes in their stereotypical behaviors during their depressive episodes. It is, therefore, likely that the diagnostic markers commonly used for investigating depressive symptoms are not indicative of people with ASD [98]. 

This could be especially true for high-functioning individuals with ASD [102] and especially for those with a camouflaging attitude [103]. Among behavioral or social markers of suicidal ideation or suicide risk, camouflage (the attempt to hide autism or to overlay some of its symptoms) was found to be one of the most highly significant. Adults with autism who blend in themselves are eight times more likely to get hurt than those who do not [104]. Consequences of camouflaging include exhaustion and threats to self-perception [105]. Moreover, the effort of camouflage contributes to anxiety and depression, therefore increasing the risk of suicide [103,104]. Depression is also more common in women with ASD, and this is likely linked to a later diagnosis, the sense of not being understood by others, and the more typical tendency to “camouflage” symptoms [106].

Furthermore, it has been shown that anxiety in children with ASD, with preserved cognitive functions, is a prognostic factor for clinical depressive symptoms and suicidal ideation [107]. Most of the authors underline the importance of a comprehensive assessment to identify anxiety symptoms [98] in order to be able to provide appropriate treatment, especially considering that anxiety in adolescence represents a significant risk factor for psychiatric illnesses and suicidal ideation.

Among the disorders potentially associated with ASD, attention has been paid to the schizophrenia spectrum disorder, including the so-called “psychotic experiences” [108], as risk factors for suicidality. Higher ASD traits and positive psychotic symptoms seem to be associated with increased depression, hopelessness, and suicidality [109]. Consequently, all ASD subjects with psychotic experiences should be screened for depression and suicidality with the purpose of re-modulating therapeutic interventions if necessary.

In general, studies have examined potential mechanisms underlying thoughts of self-harm and depression in ASD, demonstrating that the perception of loneliness and the low quality of social support received contribute to suicidal ideation and can be considered risk factors [110].

In conclusion, in accordance with the interpersonal theory of suicide [13], ASD status among children is significantly associated with a higher probability of experiencing adverse childhood experiences (ACEs) that potentially compromise their chances for physical and behavioral health outcomes [111]. Negative experiences, such as childhood maltreatment and the resulting perception of being unwanted, may elevate the risk of perception of burdensomeness that, in turn, increases the risk of suicide.

Early diagnosis of depression and awareness of the risk of suicidal thoughts or attempts in individuals with ASD are fundamental factors in preventing suicidality and providing adequate psychological support [112]. Recommending social interventions that address loneliness could lead to better mental health outcomes in ASD people [110]. Even preventing isolation from others and bullying consequences, as potential environmental intermediary factors, could be a target for the prevention of suicide [99].

### 3.5. Specific Learning Disorder and Suicide 

Specific learning disorder (SLD) affects specific skills, including word reading accuracy, spelling, grammar, and calculation. When not promptly diagnosed, difficulties with these skills may interfere with learning, academic progression, and professional success and may impact everyday activities and social interactions [18,113]. The prevalence of SLDs has been estimated to range from 5% to 17.5% [114]. SLDs are highly heritable [115] with multifactorial etiology [116].

There is a lack of studies focused on suicidality in children and adolescents with SLD. However, it is well known that children with SLD are at risk of developing psychiatric disorders or behavioral and psychological problems [117,118,119,120]. Struggling with learning to read and write is a source of frustration, and children and adolescents with SLD more frequently than their peers present low self-esteem, anxiety, and depressive symptoms related to school failure [119,120,121,122,123,124]. Internalizing disorders have frequently been described in meta-analysis studies [125,126], suggesting that, compared to typical readers, poor readers are at moderate risk for experiencing internalizing problems, such as anxiety and depression [127]. Conversely, positive academic self-concept seems to be associated with a lower reporting of suicidal ideation [128].

The same reports describe more frequent suicidal ideation or attempts and school dropout in students with reading difficulties than in skilled reading students [129]. In the 2012 nationally representative Canadian Community Health Survey (n = 21,744 people), the prevalence of lifetime suicide attempts among those with SLD was found significantly higher than those without (11.1% vs. 2.7%; *p* < 0.001) [130]. In this study, only the “adverse childhood experiences” (ACEs) significantly accounted for the association between SLD and suicidal attempts. However, the association between SLD and psychological symptoms is unclear due to the lack of longitudinal studies. A recent study using longitudinal data [117] investigated the relationship between dyslexia and children’s mental health in China. The results of this study showed the negative impact of SLD on children’s mental health, highlighting a strong association between dyslexia and depressive symptoms.

Moreover, according to other studies, people with SLD can develop post-traumatic stress disorder (PTSD) related to school-based trauma [131]. People with SLD have been seen to be at risk of developing negative emotional coping strategies with the development of self-harm behaviors and suicide attempts [132]. Suicidality can also be related to learned helplessness, a condition characterized by a sense of powerlessness due to traumatic events or a persistent failure to succeed [133]. It is thought to be one of the underlying causes of depression, very common in people with SLD [132]. 

In summary, dyslexia and related school trauma predispose the onset of dysfunctional coping strategies with increased risk, compared to normal readers, of developing internalizing disorders such as depression, self-harm behaviors, and suicidality. Indeed, the longevity of school trauma can have important effects on the future fulfillment of affected children [132]. For this reason, during the school career of children with dyslexia, we should focus on the prevention of these unfavorable outcomes, identifying early the risk indicators of emotional distress related to repeated failures and school frustrations.

### 3.6. Tic and Tourette Disorder and Suicide

According to DSM-5 [18], Tourette disorder (TD) is a childhood-onset neurodevelopmental disorder characterized by multiple motor and vocal tics present for at least one year. The prevalence of the disorder is up to 1% of the population with a gender ratio of 3 M:1 F [134,135]. A diagnosis of persistent or chronic motor/vocal tic disorder (CTD) is given when only motor or vocal tics are present [18].

The risk of suicide is considerably high in individuals with TD/CTD. A large cohort study of 7736 individuals with TD/CTD from the Swedish National Patient Register, compared with control subjects, reported an increased risk of both dying by suicide (odds ratio: 4.39; 95% confidence interval [CI]: 2.89–6.67) and attempting suicide (odds ratio: 3.86; 95% CI: 3.50–4.26) [136]. Suicide attempts such as hanging, strangling, or suffocation are frequently reported among TS/CTD patients [136]. TD/CTD also have the propensity to utilize self-poisoning techniques, maybe due to the prompt availability of psychotropic medication [136]. A recent systematic literature review [137] yielding 20 cohort studies suitable for quantitative synthesis reported self-injurious behaviors in 35% of patients with TS. The authors highlighted a significant correlation between obsessive-compulsive behaviors and self-injurious behaviors in individuals with TS.

Overall literature data highlight that the majority of individuals with TS present concomitant behavioral problems, most commonly obsessive-compulsive disorder (OCD) and ADHD [138,139]. Even though anxiety disorders and explosive outbursts are not formally included in the diagnosis of TS/CTD, they are prevalent in individuals with TD and could heighten the risks of both peer victimization and suicide [140,141]. The persistence of tics beyond young adulthood and a previous suicide attempt is the strongest predictor of death by suicide in individuals with TS or CTD [136,142].

Children with TS/CTD and associated psychiatric conditions are at an increased risk of being involved in bullying behaviors [136,143]. Since bullying and peer victimization are strong risk factors for later suicidality [144,145], specific bullying screening and suicide prevention are strongly suggested in this population [140]. From a psychopathological perspective, children with TS/CTD who experience peer victimization (‘‘victims’’) show greater tic symptoms (increased total tic severity and premonitory urge intensity), internalizing symptoms (i.e., depression, anxiety), explosive outbursts, and poorer psychosocial functioning and quality of life [140].

Another strong risk factor is the high level of drug and alcohol abuse, commonly described in TS/CTD patients who have attempted suicide [136]. In association, social isolation, family history of autoimmune diseases, substance use, and an unhealthy lifestyle have been considered as conditions linked to suicidality in TD/CTD individuals [141]. 

Finally, stress and emotion dysregulation seem to be significantly related to both tic expression and severity of comorbidities [146]. Therefore, the potential mediating role of emotion regulation in SSBs in this population needs to be further investigated.

### 3.7. Interventions 

Extensive treatment of the intervention models and tools for SSBs is beyond the scope of this review. No specific studies on the effectiveness of both psychotherapeutic treatment and psychopharmacological agents in cohorts of subjects with NDDs at risk for suicide are available. Therefore, the therapeutic approach is currently based on the general principles for intervention in the pediatric population. Some empirical support is available for the efficacy of psychotherapeutic treatment and psychopharmacological agents. Among psychotherapeutic approaches, “Integrated Cognitive-Behavioral Therapy (I-CBT)”, the “Multisystemic Therapy (MST)”, the “Mentalization-Based Treatment for Adolescents (MBT-A)”, “Developmental Group Psychotherapy (DGP)”, the “Resourceful Adolescent Parent Program (RAP-P)”, and the “Dialectical Behavior Therapy for Adolescents (DBT-A)” have received attention [147]. The main pediatric study on antidepressants and mood stabilizers for suicidal behavior is the Treatment of Adolescent Suicide Attempters study (TASA) [148], which had an uncontrolled, open treatment design. Medication was found to be no more effective than psychotherapy or combination treatment in reducing suicidal outcomes [149]. Strong evidence in favor of the anti-suicidal properties of Lithium treatment has been collected since the early 1990s [150,151], even though there is a lack of controlled studies in children and adolescents. 

In 2004, the FDA issued a warning on a possible increased risk of suicide in children and adolescents taking antidepressants, prompting numerous studies on the safety of SSRI use in the pediatric population [152,153,154]. A recent meta-analysis [155] confirmed that antidepressant exposure is associated with an increased suicidal risk in the pediatric population. Nevertheless, it has been demonstrated that the consequential worry in using antidepressants led to a drop of 25% in rates of both diagnosis and treatment of depression by pediatric and non-pediatric primary care physicians [156]. In parallel, an increase of 25% in the completed suicide rate in children and adults has been described in the last two decades, feasibly as a consequence of a sort of “barrier” to the pharmacological treatment of depression [157]. Overall, the available literature seems to agree on the benefit of the use of medication because of a higher rate of relief from depressive symptoms than not using any medication. It should be underlined that untreated depression remains one of the main risk factors for suicide and that the effectiveness of antidepressants is well established. Therefore, it is recommended to use the SSRI, even though cautiously, especially in young patients at risk for suicidality. The AACAP Parents Medical Guide Workgroup [158] recommends that parents and caregivers manage the increased risk for suicidal thoughts by careful monitoring, the development of a safety plan, and the combination of medication with psychotherapy.

Current empirically supported treatments are not entirely satisfactory in alleviating depression in the pediatric population. Some authors suggest that the current treatments fail to alter pervasive negative self-representations and affect dysregulation typically associated with chronic depressive symptoms. New investigations on the neurofeedback effects on positive self-processing as a protective condition against chronic depression have been conducted [159,160]. Particularly during adolescence, neurofeedback could represent a potential alternative treatment for depression due to its ability to engage the limbic functional circuitry (i.e., targeting the right amygdala, hippocampus, and anterior cingulate cortex), underpinning the depressive “rumination”, the autobiographical memories, and the implicit emotion regulation circuitry. Protective effects of neurofeedback have also been recently reported in youth with NDDs such as ADHD [161].

Some exploratory studies have been conducted to examine the effect of repetitive transcranial magnetic stimulation (TMS) on suicidal ideation and depressive symptoms in adolescents have been conducted. Preliminary findings suggested that TMS can mitigate suicidal ideation in adolescents through the treatment and improvement of depressive symptom severity [162]. Very recently, a repetitive TMS (rTMS) study aimed to treat adolescents with NDD demonstrated changes in brain activities and functional connectivity [163]. In particular, changes in the left subgenual anterior cingulate cortex and in the right middle cingulate cortex seem to be associated with an improvement in depressive symptoms, suggesting possible neural targets for further clinical trials.

Finally, promising interventions and routes for youth suicide prevention have been developed through the use of new technologies [164]. A recent systematic review [165] showed that telemedicine is the most adopted tool. However, evidence has also been found for mobile applications focused on the screening of depressive symptoms and suicidal ideation, as well as for clinical monitoring through the use of text messages. The authors observed that the technological tools for suicide prevention are well accepted and tolerated by youths. Nevertheless, only a few data are still available on the efficacy of technology mediated interventions.

None of these interventions have provided data on their effectiveness in specific cohorts of subjects with NDDs. The peculiar cognitive and/or behavioral profiles of the youth with NDD would modify the response to both pharmacological and non-pharmacological treatments. The implications of these differences for the management of suicide risk in this vulnerable population should be analyzed and discussed by pediatric mental health specialists.

## 4. Discussion

Our narrative review aimed to examine the literature contributions to the risk factors for self-harm behavior and suicide in children and adolescents with NDDs. A comprehensive overview of epidemiology and risk factors of SSBs has been described for all six neurodevelopmental disorders. More extensive literature has been found on children and adolescents with ADHD, ASD, and Tourette disorder, highlighting both general risk factors (e.g., depression, ACE, emotion dysregulation) and specific risk factors (e.g., camouflage in ASD population, the risk to be involved in bullying behaviors in TD/CTD) associated with suicidality for young people with these disorders. We also found some gender differences, such as the relatively higher prevalence of suicide attempts in females than in males with ADHD or a relative risk rate higher in females with ADHD compared to the general population and slightly higher than in males with ADHD.

Despite the paucity of research on the other NNDs, a clear and urgent need emerged to raise awareness for suicide as a genuine risk for people with IDs [29], communication disorders [39], and specific learning disorders [132]. This second group of NDDs probably fail to obtain the attention they deserve in terms of risk for suicide. However, many authors suggest that these conditions should be given importance due to their clinical correlations with affective disorders and, therefore, with SSDs.

A different route to suicidality for neurodevelopmental disorders, compared to the general population, has been postulated for all NDDs, although this issue has not been broadly investigated so far. Among the research topics related to suicidality in NDD, the mechanisms underlying self-harm behaviors remain largely unexamined. According to the available data, depressive symptoms are the main associated conditions with suicide both for people with neurodevelopmental disorders and the general population [166]. Nevertheless, screening for depression and suicide risk in children and adolescents with NDDs is a tough challenge, and the risk of misrecognizing the depressive symptoms and the red flags for suicide risk is very high in children and adolescents with NDDs. For instance, depressive disorders are frequently mis- and underdiagnosed in individuals with ASD [167].

Nevertheless, screening for depression and suicide risk in children and adolescents with NDD is a challenging task, requiring appropriate tools and strategies that are currently unavailable. Therefore, the risk of misrecognizing the depressive symptoms and red flags of suicide risk is very high in children and adolescents with NDD. 

Among the factors strongly associated with suicidality, ED is one of the most extensively studied in NDDs. ED has been described as an independent factor of increased vulnerability to suicidal ideation among adolescents and young adults with mood disorders [168]. Children and adolescents with NDDs are more prone than the general pediatric population to experience exaggerated emotional states and difficulties in regulating emotions. Thus, ED has been conceptualized as a transdiagnostic feature in young individual NNDs [169]. According to Linehan’s biosocial theory [170], ED should be considered a crucial factor in maintaining suicidality, since suicidal ideation has been viewed as a strategy aimed to reduce or avoid an overwhelming emotional status.

A second crucial factor influencing suicidality in NDDs seems to be the ACE burden. Adolescents with a history of ACEs, which include a broad range of conditions, such as family violence, household mental illness or alcohol/drug problems, parental divorce/separation, parental death, household poverty, and parental incarceration, are at an increased risk of depression [171] and suicide [123]. Conversely, significant and positive associations between ACEs and neurodevelopmental and behavioral disorders have been found, even after adjusting for other covariate factors [172]. In particular, the association between prenatal alcohol and drug exposure, ACEs, and diagnosed neurodevelopmental disorders has been broadly studied [173,174]. Therefore, the burden of ACEs could be one of the main mediators of the association between NDDs and suicidality. 

Several classes of risk factors have been identified in community-based studies of the pediatric population, with the strongest risk factors represented by current suicidal ideation, past history of attempts, or recent attempts by a friend or family member [175]. To the best of our knowledge, no specific studies on cohorts of individuals with NDDs have been conducted with the aim of identifying specific risk factors for SSBs. The diagram in Figure 2 was built upon the studies included in this review and represents the risk factors for suicidal spectrum behaviors shared across individuals with all the different forms of NDDs. 

Both emotion dysregulation (as an individual factor) and ACE (as an environmental factor) have been mentioned as well-established risk factors for suicidality in all NNDs. In a theoretical algorithm, ED and ACE can lead to SSBs directly or through depressive spectrum disorders. Instead, Table 1 describes some of the specific risk factors more frequently identified in each of the six neurodevelopmental disorders. In our opinion, further research on this topic should start from the awareness that children and adolescents with NDDs have peculiar neuropsychological and psychopathological characteristics and need specific pathways for the identification, prevention, and treatment of SSBs.

A strong association between emotional/behavioral dysregulation and suicidal behaviors has been consistently described as an independent risk factor across both externalizing and internalizing disorders [176]. Within the context of emotional dysregulation (ED), irritability may play a significant role in suicide and suicidal behaviors. Emotional and behavioral disturbances associated with irritability have been frequently reported during the period preceding suicidal behavior [177,178]. A systematic review designed to understand the relationship between irritability and suicide [179] showed that teens with self-reported high levels of “anger” (a term used as an equivalent to irritability) were more likely to report suicidal ideation. Interestingly, a 35-year longitudinal study [180] demonstrated that irritability, as rated by parents, represents a significant risk factor for suicidal behavior later in adulthood, beyond the associations with other adult disorders. According to the Interpersonal-Psychological Theory of Suicide (IPTS), three causal factors are crucial for suicide: the “perceived burdensomeness”, the “failed belongingness”, and the “acquired capability for suicide” (ACS). ACS is defined as heightened fearlessness toward death and increased pain tolerance, enabling the acquisition of the capability to overcome the basic self-preservation instinct. Emotion dysregulation is a crucial variable within the framework of the IPTS [181]. Individuals more prone to be easily overwhelmed by negative emotions have an increased risk for both suicidal desire and ACS [12]. Moreover, a reciprocal relationship between ED and suicidal ideation has been described since the lack of ability to regulate emotions may trigger thoughts of suicide as a strategy to alleviate emotional suffering. Conversely, ruminating thoughts about death can alter the self-efficacy of regulation skills [182]. The same reciprocal pathways have been described for non-suicidal self-injury (NSSI), which can be regarded as a maladaptive coping strategy for negative and high emotional states resulting from low regulation abilities [183]. 

Even though most of the reported research has not been specifically designed for individuals with NDDs, it consistently emphasizes emotional dysregulation (ED) as a key transdiagnostic dimension in this population. Thus, we must mention the assessment of emotion dysregulation since it represents a crucial factor in suicidal behaviors. A large body of studies insists on the Child Behavior Checklist (CBCL)–Dysregulation Profile (DP) [184], which has been empirically designed as a mixed phenotype of severe dysregulation arising from elevated scores on CBCL subscales of anxiety/depression, attention deficit/hyperactivity, and aggression. The CBCL-DP has been linked to adolescent self-harm, suicidal ideation, and suicidal behaviors [185,186]. Another widely used self-report measure of ED is the Difficulties in Emotion Regulation Scale (DERS) [187], originally designed for adults. Nevertheless, it has been used to study suicidal ideation and attempts also in adolescent inpatients [188]. The findings revealed that the limited ability to access emotion regulation strategies was significantly associated with suicidal ideation. Moreover, a history of past suicide attempts was positively correlated with a higher degree of affect dysregulation.

To this end, a crucial question is whether routine screening of all patients with NDDs referred to a mental health care service would be a suitable practice for the prevention of suicide risk. A general agreement among researchers has been reached on the opportunity to consider suicidality in NDDs as a serious and complex issue requiring the implementation of protective protocols. According to the American Association of Pediatrics and the National Strategy for Suicide Prevention, screening for suicide risk is a method of suicide prevention for the general population [189]. In the last two decades, a lot of effort has been put into finding suicide prevention strategies and screening instruments specific to the young population, different from adult suicide prevention strategies [190]. Concerning the use of questionnaires to evaluate suicide risk, there is a general agreement that evaluations based on interviews and questionnaires may be flawed by cognitive bias or the simply fleeting nature of emotions. However, recent data suggest that universal screenings, such as the Patient Health Questionnaire (PHQ-9) screening instrument, largely used in the adult population [191], may also increase identification and treatment initiation for adolescents at risk for suicide [192]. Is this also true for children and adolescents with NDDs? 

Widespread screening for suicide ideation across pediatric healthcare settings has been proposed as secondary prevention (e.g., [169]). Can the tools used for risk identification commonly used in the general pediatric population function effectively in cohorts of individuals with NDDs? The Ask Suicide-Screening Questions set is one of the most extensively used and highly sensitive instruments used to detect suicide risk in youths [32]. Does this instrument need to be modified in order to study this fragile population? Finally, some of these screening procedures involve the nursing staff in the application of validated screening tools and, therefore, in the identification of at-risk individuals. Is it feasible for individuals with NDDs as well? Unfortunately, the current literature does not yet provide answers to these questions. However, recent research [16] describes the results of a 6 month study of a population with NDDs (aged 8 to 18 years) screened by nurses using the “Ask Suicide-Screening Questions” tool during triage. A high percentage (73.3%) of screenings were completed for 3854 individuals, with 261 positive screenings noted (6.8%). Among 187 children (38.5%) with positive screenings, 72 were recognized and referred to outpatient mental health referrals. Seven (2.5%) of them required acute psychiatric treatment. Compared to all other NDDs, ASD was the condition with higher rates of positive screenings. Even though this study needs to be replicated, it represents the first demonstration that a routine screening performed by a nursing staff through the “Ask Suicide-Screening Questions” aimed to identify the suicide risk among children with NDDs that was both feasible and useful for referring the at-risk subjects to the mental health care services.

Beyond the combination of structured and unstructured interviews with clinical questionnaires, the assessment of neuropsychological abilities, such as attention and impulsiveness by a GoNoGo task [193] or the preference for negative stimuli in a dot-probe task [194], has been recently proposed as a candidate for assessing suicide-specific biases in adults’ population. Previous studies have identified physiological markers that may help identify suicidal patients, including changes in heart rate or breathing as a response to the dysregulation of the interplay between parasympathetic and sympathetic branches of the autonomous nervous system [195,196]. An inverse correlation between parasympathetic activity measured through the high cardiac frequency component and suicidality has also been recently found in an acutely suicidal population of adolescents [197]. Similarly, the quantified electroencephalogram (QEEG) has been proposed as an objective measure of suicide risk. Some studies showed a positive correlation between suicide attempts and elevated high gamma power in the frontal and temporal lobes [198]. Very recent data on the association between non-suicidal self-injury (NSSI) and quantified electroencephalogram (QEEG) in adolescents and young adults with depression [199] and on resting-state EEG coherence differences in adolescents with depression and suicidal ideation or behaviors [200] seem to be very promising. Again, the question of whether these biomarkers are helpful for risk identification probability in the general adolescent population or that they may work properly also in cohorts of individuals with NDDs is still unanswered.

Finally, a plethora of genetic, familial, and twin studies have described a genetic contribution to suicidal thoughts and behaviors [201]. For instance, it has been shown that exposure to adverse experiences combined with genetic and epigenetic vulnerabilities may lead to a chronic impairment of HPA axis normal functioning, that, in turn, is a neurobiological correlation of emotion dysregulation [202,203,204]. A more recent genome-wide association study described genetic risk variants associated with suicide death and suicidal behavior. A meta-analysis involving 8315 cases vs. 256,478 psychiatric or population controls [205] found one locus in neuroligin 1 (NLGN1), passing the genome-wide significance threshold for suicide death. Interestingly, NLGN1 encodes a member of a family of neuronal cell surface proteins, acting as splice site-specific ligands for beta-neurexins and, therefore, being involved in synaptogenesis. From an ontogenetic point of view, this evidence suggests a robust link with NDDs, since this family of genes is strongly implicated in the pathogenesis of NDDs [206].

### Limitations

Very few studies of SSBs have examined the role of variables apart from psychiatric diagnoses in the pediatric population with NDDs. Furthermore, we did not find satisfactory information about the relationship between suicidal behaviors and temperament constructs in subjects with NDDs. In our opinion, the temperamental variable has a significant impact on the NDD phenotype and, inevitably, on the SSBs. They should be studied with the purpose of enhancing awareness of the potential risks and protective factors of SSBs. Another limitation lies in the lack of standardized instruments specifically designed to assess suicidality in most of the studies. Therefore, we summarized the available data in a narrative form. To address these gaps, systematic literature reviews and, hopefully, metanalysis should be performed as more quantifiable information becomes available. 

## 5. Conclusions

Our work provides conceptual grounds for researchers to improve the recognition of the risk factors of SSBs as the first step in the context of a prevention plan for children and adolescents with NDDs. The challenges and implications of screening for suicide risk in this vulnerable population, as well as the role of pharmacological and non-pharmacological treatments, should be seriously debated by pediatric mental health specialists. This review intended to underline the specific needs of this population, highlighting the possibility that the evidence on suicidality from research on the general pediatric population is not *tout court* valid for children and adolescents with NDDs as well.

The knowledge of the specific routes for suicidality among children and adolescents with NDDs should be increased with the purpose of identifying the best practices aimed to drive the developmental trajectory of these young people with NDDs far from the conditions underlying the most dramatic outcome.

## Figures and Tables

**Figure 1 jcm-13-01627-f001:**
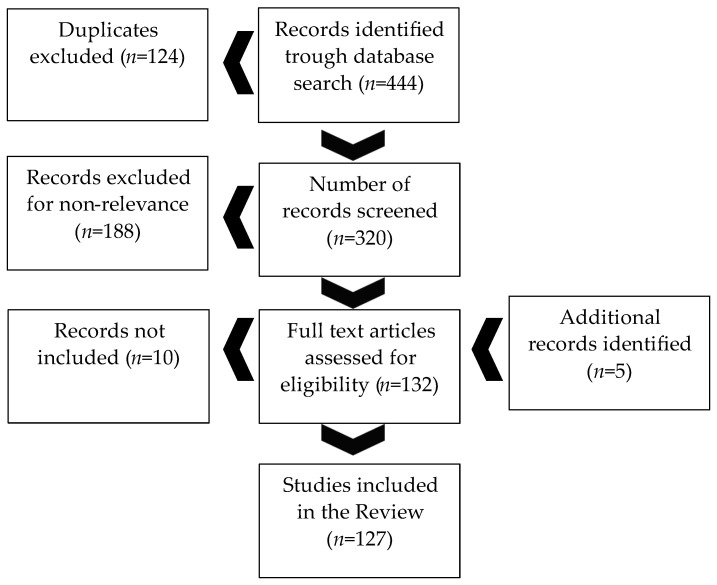
Review process.

**Figure 2 jcm-13-01627-f002:**
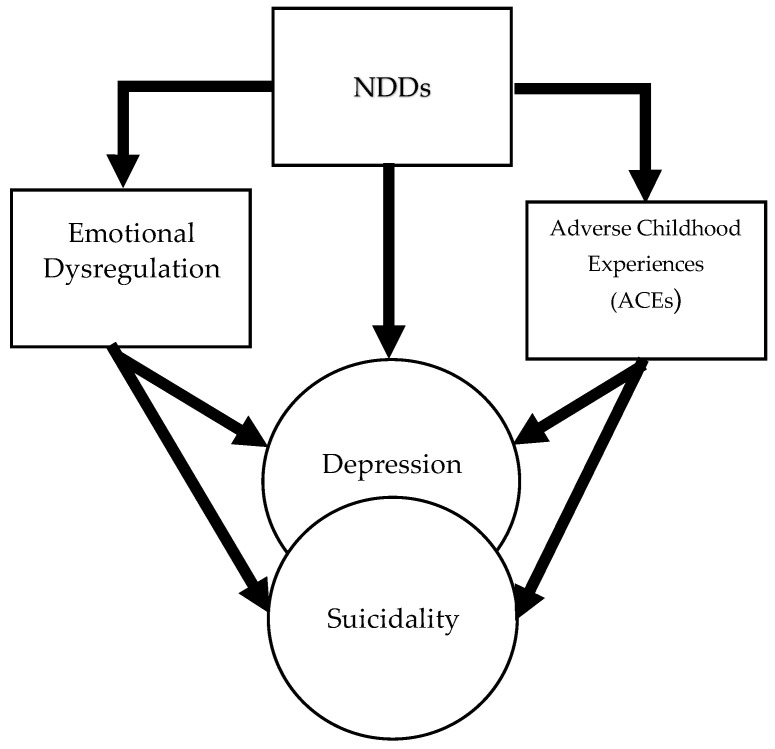
Shared risk factors for suicidal spectrum behaviors (SSBs) in NDDs. The model intends to focus on the main routes to suicide in subjects with NDDs, over and above the depression, as suggested by the lowest two arrows going directly from emotion dysregulation (ED) and adverse childhood experiences (ACEs) to suicide. However, this is a necessarily reductive model and more and different routes should be taken into consideration, as elucidated in the discussion.

**Table 1 jcm-13-01627-t001:** Specific risk factors for suicidal spectrum behaviors (SSBs) for each neurodevelopmental disorder (NDD).

Intellectual Disability	Communication Disorders	Attention-Deficit/Hyperactivity Disorder	Autism Spectrum Disorder	Specific Learning Disorders	Tic and Tourette Disorder
Mild level of intellectual disability	Social anxiety and isolation	Executive dysfunction and impulsivity	High intellectual functioning	Negative-academic self-concept	OCD
Depression, anxiety, and psychosis	Sense of loneliness	ODD, conduct disorder, and substance use	Camouflaging	Learned helplessness	Bullying and alcohol abuse

## Data Availability

The data presented in this study are available upon request from the corresponding author.

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
