# Peer review of "Neurodevelopmental Disorders and Suicide: A Narrative Review"

_jcm, 2024, doi:10.3390/jcm13061627_

Round 1

Reviewer 1 Report

Comments and Suggestions for Authors

Thank you for the opportunity to review this interesting paper. I appreciate the tremendous effort put into reading all the papers selected and putting together information that can potentially be of use to both mental health professionals and researchers in the field.

While I value the information provided and have enjoyed reading through the manuscript, I believe there are some major issues and flaws that need to be addressed (please see below). Further, I also recommend consulting an English-speaking scientific editor to address major imperfections regarding grammar, syntax, and punctuation.

Abstract

It's hard to read and grasp. It needs to be revised, improving grammar, syntax, and punctuation.

Introduction

1) Lines [33-34]: "Each of these deaths is a tragedy as they could be avoided with the appropriate prevention strategies".

Not sure of the consequentiality between tragedy and the fact that each death could have been avoided. I would suggest that the fact that each death is a tragedy is more related to the scars and/or social consequences (including stigma) that it may leave in the family.

Whether a particular death could have been avoided needs to be discussed separately. So this sentence should be elaborated/reworked or deleted.

2) Line 41: Please replace SSBs with SBB

3) Line 48: developmental disorders (NDD): The abbreviation doesn't add up. Do you mean neurodevelopmental disorders?

4) Line 51: Behaviour vs behavior. Please use either British or American English, based on the editorial guidelines of the Journal.

5) Lines [50-52]: ...that is the intentional destruction of body tissue without suicidal intent, su- 50 icidal attempt (SA) that is non-fatal self-injurious behavior with the intent to die, until 51 suicide death [4]:

This is sentence is very hard to grasp. Further, "intentional destruction of body tissue" sounds a little odd. Please find a more clinical expression, possibly more commonly used in the suicide literature. 

6) Lines [53-63]. The authors didn't mention that psychotic disorders significantly increase suicide risk in children and adolescents:
https://www.sciencedirect.com/science/article/abs/pii/S0920996421002826

Given the role of prenatal and perinatal risk factors in schizophrenia, I would recommend adding a sentence or two on the topic.

Methods
1) Lines [94-95]: This review helps better delineate the state-of-the-art main research findings about the relationship of suicide in NDDs. 

And why is that?

2) Lines [95-101]: Please provide the exact search string(s) used for each of the libraries used.

Results
3) Lines [124-125]: It is estimated that 1-3% of the population, or seven to eight million Americans, have 124 an Intellectual Disability [16].

Why are these data relevant? Is the goal of the paper to discuss suicide in the US? Also, Americans from where exactly? the US? Canada? Central and South America??

4) There is no need to add "...and Suicide" to each paragraph

5) Line [203]: "temporal distancing for emotion regulation". 
Please don't assume that all your readers know all the technical and clinical expressions in the field of psychiatry and suicide prevention.

Please make an effort to improve the readability of the present paper.

6) Line [210]: The three levels of statistical significance are:
p<0.05; p<0.01; p<0.001

So p=0.0003 should be reported as p<0.001

7) Line [221]: Attention Deficit Hyperactivity Disorder (ADHD)
The abbreviation has already been introduced

8) No need to write SSBs in its plural form, just SSB

9) Lines [335-338]: Again an example of a study conducted i the US is given, suggesting that firearms were less likely to be used. This consideration, however, is not applicable other countries where the the general population, let alone people in the ASD spectrum, have access to firearms.

10) Line 344. 54.168 or 54,168. Please check the editorial guidelines of the journal.

11) Line 347: y 8%- 12.5% should be written as 8-12.5%

12) Line 352: "...children with ASD who have less cognitive abilities"
This makes no sense. Consider rephrasing using more clinical/scientific terms

13) Line 360: "to elaborate coping strategy through the social support"
What social support?

14) Line 447: the so called “learned helplessness”. Learned helpness is a well-described emotional state, and all mental health professionals globally understand the term. So, please delete "so called".

15) Lines [498-500]: No specific studies on the effectiveness of both psychotherapeutic 498 treatment and psychopharmacological agents in cohorts of subjects with NDDs are available.

Not sure I understand. Do you mean that no studies show the efficacy of psychotherapeutic/pharmacological interventions in children/adolescents with NND? Or maybe children/adolescents with NDD at risk for suicide?

There are hundreds of studies actually but again, it's not that clear what you actually mean.

16) Lines [514-521]. The FDA gave the warning about SSRIs in the younger population because lots of evidence suggested so! So why are you now saying that it's OK to prescribe them?

How about cognitive remediation approaches like neurofeedback training? How about TMS?

Take a look at these papers for example:
https://pubmed.ncbi.nlm.nih.gov/32206963/
https://pubmed.ncbi.nlm.nih.gov/30031247/

In the last few decades research in translational medicine has made enormous progress and several technology-assisted methods in both diagnostics and treatment have become available, especially for the younger population (often based on video games and computer-human interaction).

The authors should at least provide some up-to-date information, especially considering that clinicians will be reading this paper in search of alternatives to dangerous antidepressant treatments in NND children and adolescents.

Discussion

1) Line [547]: "It would require appropriate tools and strategies, currently not available.

This is not true. Again several methods are discussed in the literature as a huge quantum leap in the evaluation of suicide risk. You can combine structured and unstructured interviews with clinical questionnaires, evaluation of psychomotor performance in GoNoGo tasks (measuring attention and impulsiveness), preference for negative stimuli in a dot-probe task, facial expression changes in response to the presentation of a battery of stimuli with variable emotional valence, HRV/EDA, and of course resting state quantitative EEG (qEEG).

This is research that needs to be mentioned in the paper to provide fresh information to the clinician. And, again, there's technology out there that allows all that at a relatively low cost for both the public and private healthcare system.

2) Figure 2. This model is way too simplistic in my opinion. Mainly because it implies that suicide is a direct consequence of depression, which is NOT always the case, as suggested by these studies for example:

https://pubmed.ncbi.nlm.nih.gov/8894062
https://pubmed.ncbi.nlm.nih.gov/31896024

Of course, the papers above are not on children and adolescents with NDD, nonetheless, they do remark that suicide is not necessarily a consequence of depression. Please discuss and improve your model.

3) Concerning the use of questionnaires to evaluate suicide risk, we know from the literature that evaluations based on interviews and questionnaires may be flawed by a cognitive bias or the simply fleeting nature of emotions.

We also know that suicidal states are typically associated with increased impulsivity and numbed emotions (a state that is necessary to win the fear of death). This is why HRV-, EDA-, and EEG/ERP-based biomarkers are badly needed to reach greater objectivity and reliability in the evaluation of real risk. The discussion should include this information and considerations in layman's terms (please avoid highly technical language).

Conclusions
Lines [632-633]: " The challenges and implications of screening for depression and suicide risk".

Again, not all suicidal persons are depressed. Please comment above and improve.

Comments on the Quality of English Language

The whole manuscript should be revised by an English-speaking scientific editor.

Author Response

​​Dear Revewer 1

​Thank you sincerely for your thoughtful review and for recognizing the effort invested in our paper. We appreciate your time and commitment to providing valuable feedback. We are grateful for your positive comments on the potential usefulness of the information for mental health professionals and researchers. Your encouragement motivates us to continue refining our work.We have carefully considered your suggestions and observations about the major issues and flaws in the manuscript. Your insights are invaluable, and we are committed to addressing each point thoroughly to enhance the overall quality of the paper.

The main changes are highlighted in yellow in the text of the manuscript. Where the manuscript was edited as a result, the revised text is included after our response, edited in red.

Furthermore, we acknowledge your recommendation to consult an English-speaking scientific editor. We will certainly take this advice seriously and seek professional assistance to rectify any grammatical, syntactical, and punctuation imperfections. Your constructive feedback is crucial to our ongoing improvement, and we look forward to presenting a revised version that aligns more closely with the standards of excellence. Once again, thank you for your time, expertise, and dedication to advancing the quality of our work.

Best regards,

The Authors

Abstract

It's hard to read and grasp. It needs to be revised, improving grammar, syntax, and punctuation.

Answer : We apologize for not paying enough attention to correct English. Our manuscript has been reviewed by a native english speaker.  

Introduction

1) Lines [33-34]: "Each of these deaths is a tragedy as they could be avoided with the appropriate prevention strategies".

Not sure of the consequentiality between tragedy and the fact that each death could have been avoided. I would suggest that the fact that each death is a tragedy is more related to the scars and/or social consequences (including stigma) that it may leave in the family.

Whether a particular death could have been avoided needs to be discussed separately. So this sentence should be elaborated/reworked or deleted.

Answer 1

Thanks for your valuable comment. We agree that the sentence is inappropriate in the context and it should be deleted.

2) Line 41: Please replace SSBs with SBB

Answer 2 Thanks, done.

3) Line 48: developmental disorders (NDD): The abbreviation doesn't add up. Do you mean neurodevelopmental disorders?

Answer 3: Thanks for the note. We corrected.

4) Line 51: Behaviour vs behavior. Please use either British or American English, based on the editorial guidelines of the Journal.

Answer 4: Thanks,  based on the editorial guidelines of the Journal, It's better if we use American English.

5) Lines [50-52]: ...that is the intentional destruction of body tissue without suicidal intent, su- 50 icidal attempt (SA) that is non-fatal self-injurious behavior with the intent to die, until 51 suicide death [4]:

This is sentence is very hard to grasp. Further, "intentional destruction of body tissue" sounds a little odd. Please find a more clinical expression, possibly more commonly used in the suicide literature. 

We have changed the sentence as follow: “Children and adolescents with developmental disorders (NDDs) show a wide range of self-destructive behaviors ranging from Non-suicidal Self-injury (NSSI, definded as the intentional damage of the body tissue without clear suicidal intent), to Suicidal Attempt (SA, defined as non-fatal self-injurious behavior with the intent to die), until suicide death”

6) Lines [53-63]. The authors didn't mention that psychotic disorders significantly increase suicide risk in children and adolescents:

Given the role of prenatal and perinatal risk factors in schizophrenia, I would recommend adding a sentence or two on the topic.

According to your very useful suggestion, we have modified the text and added a few references (Barbeito S, et al 2021; Caetano, et al, 2006; Lawrence et al, 2022).

Methods
1) Lines [94-95]: This review helps better delineate the state-of-the-art main research findings about the relationship of suicide in NDDs.  And why is that?

Answer 1: We have tried to improve the discussion consistently. We also have changed the sentence as follows: “This review intends to present the available research about suicide in individuals with NDDs”.

2) Lines [95-101]: Please provide the exact search string(s) used for each of the libraries used.

Answer 2: Thank you for the suggestion. The search was carried out by applying the following syntax: (suicide OR non suicidal self-injury) and (neurodevelopmental disorders OR intellectual disability OR ADHD OR autism OR specific learning disorder OR tic OR tourette disorder).

Results
3) Lines [124-125]: It is estimated that 1-3% of the population, or seven to eight million Americans, have 124 an Intellectual Disability [16]. Why are these data relevant? Is the goal of the paper to discuss suicide in the US? Also, Americans from where exactly? the US? Canada? Central and South America??

Answer 3: Thank you for the valuable comment, we remove this line because it’s irrelevant  for the purposes of the study.

4) There is no need to add "...and Suicide" to each paragraph

Answer 4: Thank you for the suggestion

5) Line [203]: "temporal distancing for emotion regulation". 
Please don't assume that all your readers know all the technical and clinical expressions in the field of psychiatry and suicide prevention.
Please make an effort to improve the readability of the present paper.
We fully agree. We have tried to improve it changing the phrase in text

6) Line [210]: The three levels of statistical significance are:
p<0.05; p<0.01; p<0.001
So p=0.0003 should be reported as p<0.001
Answer 6: Thank you, we corrected

7) Line [221]: Attention Deficit Hyperactivity Disorder (ADHD)
The abbreviation has already been introduced
Answer 7: Thank you for your suggestion, we have modified the line

8) No need to write SSBs in its plural form, just SSB

Answer 8: Thank you for your suggestion we have modified the line

9) Lines [335-338]: Again an example of a study conducted i the US is given, suggesting that firearms were less likely to be used. This consideration, however, is not applicable other countries where the the general population, let alone people in the ASD spectrum, have access to firearms.

Answer 9: thanks, in fact it only relates to that study and adds nothing to the discussion of the topic.

10) Line 344. 54.168 or 54,168. Please check the editorial guidelines of the journal.

Answer 10: Thanks, done.

11) Line 347: y 8%- 12.5% should be written as 8-12.5%

Answer 11: Thank you for the valuable comment

12) Line 352: "...children with ASD who have less cognitive abilities "
This makes no sense. Consider rephrasing using more clinical/scientific terms

We apologize for the imprecise terms. We have modified the sentence. 

13) Line 360: "to elaborate coping strategy through the social support"
What social support?

It doesn’t make sense: We have shortened the sentence removing “social support”

14) Line 447: the so called “learned helplessness”. Learned helpness is a well-described emotional state, and all mental health professionals globally understand the term. So, please delete "so called".

Answer 14: Thank you for the suggestion

15) Lines [498-500]: No specific studies on the effectiveness of both psychotherapeutic 498 treatment and psychopharmacological agents in cohorts of subjects with NDDs are available.

Not sure I understand. Do you mean that no studies show the efficacy of psychotherapeutic/pharmacological interventions in children/adolescents with NND? Or maybe children/adolescents with NDD at risk for suicide?
There are hundreds of studies actually but again, it's not that clear what you actually mean.

Yes, we are sorry. It was not clear. We have added “at risk for suicide”. We would like to underline the specific needs of this population and the possibility that the evidence from the research on the general pediatric population is not valid for children and adolescents with NDDs too.

Thus we have added this sentence in the conclusion section.

16) Lines [514-521]. The FDA gave the warning about SSRIs in the younger population because lots of evidence suggested so! So why are you now saying that it's OK to prescribe them?

Thank you very much for your prompt to better clarify this topic. We have added a more extensive comment in the manuscript

How about cognitive remediation approaches like neurofeedback training? How about TMS?

Take a look at these papers for example:
https://pubmed.ncbi.nlm.nih.gov/32206963/
https://pubmed.ncbi.nlm.nih.gov/30031247/

In the last few decades research in translational medicine has made enormous progress and several technology-assisted methods in both diagnostics and treatment have become available, especially for the younger population (often based on video games and computer-human interaction).

The authors should at least provide some up-to-date information, especially considering that clinicians will be reading this paper in search of alternatives to dangerous antidepressant treatments in NND children and adolescents.

We are very grateful for the several suggestions that considerably improved our manuscript. We have integrated all the recommended issues and references into the treatment section.

Discussion

1) Line [547]: "It would require appropriate tools and strategies, currently not available.

This is not true. Again several methods are discussed in the literature as a huge quantum leap in the evaluation of suicide risk. You can combine structured and unstructured interviews with clinical questionnaires, evaluation of psychomotor performance in GoNoGo tasks (measuring attention and impulsiveness), preference for negative stimuli in a dot-probe task, facial expression changes in response to the presentation of a battery of stimuli with variable emotional valence, HRV/EDA, and of course resting state quantitative EEG (qEEG).

This is research that needs to be mentioned in the paper to provide fresh information to the clinician. And, again, there's technology out there that allows all that at a relatively low cost for both the public and private healthcare system.

2) Figure 2. This model is way too simplistic in my opinion. Mainly because it implies that suicide is a direct consequence of depression, which is NOT always the case, as suggested by these studies for example:

https://pubmed.ncbi.nlm.nih.gov/8894062
https://pubmed.ncbi.nlm.nih.gov/31896024

Of course, the papers above are not on children and adolescents with NDD, nonetheless, they do remark that suicide is not necessarily a consequence of depression. Please discuss and improve your model.

We are aware that the proposed model appears too simplistic.. It just intends to focus the main routes to suicide in subjects with NDDs, over and above the depression, as suggest by the lowest two arrows going directly from ED and ACEs to suicide. However, we found your observation very appropriate. Thus we have tried to better clarify the purpose of the model and we underlined its necessarily reductive approach. We sincerely appreciated your stimulus to improve the model but we think that a simple graph may result in high impact and helpful in paying attention to the two main risk elements. Nevertheless, a larger explanation in the legend is due.

3) Concerning the use of questionnaires to evaluate suicide risk, we know from the literature that evaluations based on interviews and questionnaires may be flawed by a cognitive bias or the simply fleeting nature of emotions.

Thanks for this consideration. We hope you will excuse us for having included your very clear sentence into the manuscript (line 673-676).

We also know that suicidal states are typically associated with increased impulsivity and numbed emotions (a state that is necessary to win the fear of death). This is why HRV-, EDA-, and EEG/ERP-based biomarkers are badly needed to reach greater objectivity and reliability in the evaluation of real risk. The discussion should include this information and considerations in layman's terms (please avoid highly technical language).

According to your inspiring suggestions we have enlarged the discussion

Conclusions
Lines [632-633]: " The challenges and implications of screening for depression and suicide risk".
Again, not all suicidal persons are depressed. Please comment above and improve.

Sorry, we realized how often we made mistakes in composing sentences regarding this topic, we changed in.. “The challenges and implications of screening for suicide risk in this vulnerable population”

Reviewer 2 Report

Comments and Suggestions for Authors

This is a timely review and makes a valuable contribution to the field. The authors should have the text revised by a native English-speaker because there are many minor grammatical errors that can be easily fixed.

Comments on the Quality of English Language

Needs minor checking for grammar.

Author Response

This is a timely review and makes a valuable contribution to the field. The authors should have the text revised by a because there are many minor grammatical errors that can be easily fixed. Needs minor checking for grammar.

Answer:  Thank you so much for your appreciation of our work. We apologize for the English form and grammatical errors, the entire paper will be revised by a native English-speaker expert in the field.

Reviewer 3 Report

Comments and Suggestions for Authors

Dear Authors, 

I read with interest your work. It is well written and I do not have major concerns on its quality.

Just few points:

- I would suggest to explicit the NDD abbreviation in the abstract; 

- Line 95: there seem to be undue quotation marks at the start of the sentence: The databases PubMed and...; 

- You may consider to include in your Introduction/Discussion paragraphs a reflection on a novel approach for the study of risk factors in suicidal behavior, which is Network Analysis. I think the quality of your work would improve by mentioning this possible up-to-date strategy for a targeted prevention. 

I suggest a minor revision.

Author Response

​​Dear Reviewer,

Thanks for the corrections, we have made them in the text. You will find them highlighted in yellow.

Your suggestion to use Network Analysis is actually inspiring. We are sure that it will be feasible in the near future when more homogeneous data will be published on this issue.  

Best regards.

Round 2

Reviewer 1 Report

Comments and Suggestions for Authors

The abstract is still shaky. Here's what I'd write:

"Specific risk factors for self-harm and suicide in children and adolescents with Neurodevelopmental Disorders (NDD) may differ from those in the general population within this age range.

In the present review paper, we conducted a narrative analysis of the literature aiming to establish a connection between suicide and affective disorders in children and adolescents with NDD.

Emotion Dysregulation (ED) as an individual factor and Adverse Childhood Experiences (ACE) as environmental factors are discussed as risk factors for suicidality in all individuals with NDD.

We propose a theoretical model where ED and ACE can directly lead to self-harm or suicide, directly or indirectly by interacting with depressive spectrum disorders. Additionally, we suggest that specific risk factors are more frequently associated with each of the neurodevelopmental disorders listed in the DSM-V."

The paper has been significantly improved and I feel it now goes much in depth. I also like the new model the authors have proposed.

However, grammar, syntax, and punctuation are still VERY poor and significantly affect readability. Again, I suggest hiring a professional scientific editor to bring further improvement.

Of note, in line 724, please note that there is not just one GoNoGoTask. This is just a classification of the type of behavioral task. Please look into this.

Comments on the Quality of English Language

The paper has been improved as per my indications. However, grammar, syntax, and punctuation still need significant improvement.

Author Response

Dear Reviewer,

I want to express my gratitude for the time you dedicated to reviewing our manuscript and for the invaluable suggestions provided. I'd like to inform you that we have incorporated the abstract you proposed, which significantly contributed to bringing clarity to the objectives of our article.

Additionally, we have addressed the feedback on line 724 by substituting the definite article with an indefinite one, aiming for greater precision. Furthermore, we have conducted additional English language revisions to enhance overall coherence and language quality in the manuscript.

We highly appreciate your guidance, and we are confident that these adjustments have strengthened our article. We eagerly await any further comments or suggestions you may have.

Thank you once again for your insightful feedback.

Best regards
